# Asymmetric Measurement-Device-Independent Quantum Key Distribution through Advantage Distillation

**DOI:** 10.3390/e25081174

**Published:** 2023-08-07

**Authors:** Kailu Zhang, Jingyang Liu, Huajian Ding, Xingyu Zhou, Chunhui Zhang, Qin Wang

**Affiliations:** 1Institute of Quantum Information and Technology, Nanjing University of Posts and Telecommunications, Nanjing 210003, China; 1221014031@njupt.edu.cn (K.Z.); 2020010107@njupt.edu.cn (J.L.); 2019010103@njupt.edu.cn (H.D.); xyz@njupt.edu.cn (X.Z.); chz@njupt.edu.cn (C.Z.); 2“Broadband Wireless Communication and Sensor Network Technology” Key Lab of Ministry of Education, Nanjing University of Posts and Telecommunications, Nanjing 210003, China; 3“Telecommunication and Networks” National Engineering Research Center, Nanjing University of Posts and Telecommunications, Nanjing 210003, China

**Keywords:** quantum key distribution, asymmetric MDI-QKD, advantage distillation technology

## Abstract

Measurement-device-independent quantum key distribution (MDI-QKD) completely closes the security loopholes caused by the imperfection of devices at the detection terminal. Commonly, a symmetric MDI-QKD model is widely used in simulations and experiments. This scenario is far from a real quantum network, where the losses of channels connecting each user are quite different. To adapt such a feature, an asymmetric MDI-QKD model is proposed. How to improve the performance of asymmetric MDI-QKD also becomes an important research direction. In this work, an advantage distillation (AD) method is applied to further improve the performance of asymmetric MDI-QKD without changing the original system structure. Simulation results show that the AD method can improve the secret key rate and transmission distance, especially in the highly asymmetric cases. Therefore, this scheme will greatly promote the development of future MDI-QKD networks.

## 1. Introduction

Quantum key distribution (QKD) can unconditionally ensure the theoretical security of information transmission between two or more distant users with quantum mechanics. In the process of development from theory to practice, there are many challenges to realizing remote and secure quantum key distribution in the practical applications. With various theoretical ideas and experimental schemes being put forward, many challenges have been overcome. The BB84 protocol [1] proposed by Bennett realizes two-point communication and the Ekert91 and BBM92 protocols have been proposed successively [2,3]. Although QKD has been proven to have unconditional security in theory, imperfect devices can lead to some security loopholes that hinder the development of QKD protocols in practical applications. In practical applications, we often use weak coherent sources (WCSs) with multi-photon components, and Eve can eavesdrop with photon-number splitting (PNS) attacks [4]. Fortunately, the decoy-state method proposed [5,6] can solve PNS attacks and obtain rapid development both theoretically and experimentally [7,8,9]. Considering the imperfection of the detector, Lo firstly proposed the MDI-QKD protocol [10] which thoroughly solves the security loopholes mainly at the detection terminal. With the advantages of the MDI-QKD protocol, the MDI-QKD protocol attracts extensive attention and has been greatly studied in theory and experiments [11,12,13,14,15,16,17,18].

In previous work, the MDI-QKD was mainly studied in symmetric scenarios for simplicity. With the development of theory and technology, researchers have paid more attention to the asymmetric MDI-QKD protocol in recent years. To achieve good interference at the detection terminal, Lo proposed an asymmetric seven-intensity MDI-QKD [19], which can improve the performance of MDI-QKD in practical asymmetric structures based on the four-intensity MDI-QKD [11]. Consequently, asymmetric MDI-QKD is more suitable for the common QKD networks. However, due to its asymmetric nature, its performance is inferior to that of the original symmetric scheme. Improving the performance of asymmetric MDI-QKDs has become an urgent problem that needs to be addressed.

Inspired by the advantage distillation (AD) method [20,21,22,23], we study the principle of the method and find that the AD method can be successfully applied to the asymmetric seven-intensity MDI-QKD protocol. Compared with the original protocol, the performance of the asymmetric protocol has been significantly improved, which provides another theoretical verification that the post-processing AD method can improve the performance of the QKD protocol. This method can divide the original key string into blocks of only a few bits to achieve a high key correlation and greatly improve the protocol’s performance. The paper is organized as follows: In Section 2, we review the asymmetric seven-intensity MDI-QKD protocol and introduce the protocol with AD. The results of numerical simulations are shown in Section 3. Finally, summaries are given in Section 4.

## 2. Methods

### 2.1. Asymmetric MDI-QKD

Here, we mainly describe the process of the asymmetric seven-intensity MDI-QKD protocol, which develops from the four-intensity symmetric protocol, as follows:(1).***State preparation***. Alice (Bob) randomly prepares the signal state only in *Z* basis with $s_{A}$ ($s_{B}$), and prepares the decoy states only in *X* basis with intensities of $w_{A}$,$v_{A}$ ($w_{B}$,$v_{B}$), satisfying the formula $w_{A}$ < $v_{A}$ ($w_{B}$ < $v_{B}$). When preparing the vacuum state of intensity *o*, Alice (Bob) does not choose any base. The prepared states will be sent to Charlie to perform measurement;(2).***Measurement***. Charlie performs the Bell state measurement (BSM) after receiving the quantum states sent from Alice and Bob;(3).***Announcement***. After Alice and Bob repeat the above steps and enough counting events are recorded, Charlie publicly announces the BSM results. Subsequently, they announce the selected bases and intensities;(4).***Parameter estimation***. After finishing the quantum transmission phase, Alice and Bob can estimate the lower bound of single-photon yield $Y_{11}^{Z,L}$ and the upper bound of single-photon error rate (QBER) $e_{11}^{X,U}$ using the decoy-state technology;(5).***Post-processing***. Alice and Bob perform key reconciliation and privacy amplification on the raw key data to obtain the final secret key.

The decoupled bases are used in the asymmetric seven-intensity MDI-QKD protocol, thus the protocol can perform decoy states in the X basis only to estimate $Y_{11}^{X,L}$ and can use $Y_{11}^{Z,L}$ = $Y_{11}^{X,L}$ to obtain the single-photon yield in Z basis [11]. Then, the secret key rate can be calculated by the following formula [10,11,19]:(1)$$R=P_{s_{A}}P_{s_{B}}\left\{\left(s_{A}e^{-s_{A}}\right)\left(s_{B}e^{-s_{B}}\right)Y_{11}^{Z,L}\left[1-h\left(e_{11}^{X,U}\right)\right]-f_{e}Q_{s_{A}s_{B}}^{Z}h\left(E_{s_{A}s_{B}}^{Z}\right)\right\},$$
where $P_{s_{A}}$ and $P_{s_{B}}$ each correspond to the probability that Alice or Bob emits the signal states of $s_{A}$ or $s_{B}$, respectively. $Q_{s_{A}s_{B}}^{Z}$ and $E_{s_{A}s_{B}}^{Z}$ are the gain and QBER in the Z basis, $Y_{11}^{X,L}\left(e_{11}^{X,U}\right)$ is the lower (upper) bound of single-photon yield (QBER), which can be estimated from the decoy-state technology, h(x) is the binary entropy function, and $f_{e}$ is the error correction efficiency.

Based on the asymmetric seven-intensity MDI-QKD protocol above, the performance can be further improved by optimization techniques such as joint estimations and collective constraints [11]. Referring to the joint estimations method, the common part H is extracted from the following two parameters $Y_{11}^{X,L}$, $e_{11}^{X,U}$ to optimize the key rate. $e_{11}^{Z,U}$ is used in the following subsection. $Y_{11}^{X,L}$ is a piecewise function where $P_{v_{A}}^{1}P_{w_{A}}^{2}P_{w_{B}}^{1}P_{v_{B}}^{2}$ < $P_{w_{A}}^{1}P_{v_{A}}^{2}P_{v_{B}}^{1}P_{w_{B}}^{2}$[19,24]. These parameters $Y_{11}^{X,L}$, $e_{11}^{X,U}$ and $e_{11}^{Z,U}$ can be estimated accurately by the decoy-state technology in the original MDI-QKD protocol [10,11]. The following formulas can estimate these parameters which lead to a much higher rate in distilling the secure final key:(2)$$\begin{array}{cc} \; & Y_{11}^{X,L}=Y_{11}^{X,e}=\frac{P_{v_{A}}^{1}P_{v_{B}}^{2}Q_{w_{A}w_{B}}+P_{w_{A}}^{1}P_{w_{B}}^{2}P_{v_{A}}^{0}Q_{ow_{B}}+P_{w_{A}}^{1}P_{w_{B}}^{2}P_{v_{B}}^{0}Q_{v_{A}o}}{P_{w_{A}}^{1}P_{v_{A}}^{1}\left(P_{w_{B}}^{1}P_{v_{B}}^{2}-P_{w_{B}}^{2}P_{v_{B}}^{1}\right)}\\ \; & -\frac{P_{w_{A}}^{1}P_{w_{B}}^{2}Q_{v_{A}v_{B}}+P_{w_{A}}^{1}P_{w_{B}}^{2}P_{v_{A}}^{0}P_{v_{B}}^{0}Q_{oo}}{P_{w_{A}}^{1}P_{v_{A}}^{1}\left(P_{w_{B}}^{1}P_{v_{B}}^{2}-P_{w_{B}}^{2}P_{v_{B}}^{1}\right)}-\frac{P_{v_{A}}^{1}P_{v_{B}}^{2}H}{P_{w_{A}}^{1}P_{v_{A}}^{1}\left(P_{w_{B}}^{1}P_{v_{B}}^{2}-P_{w_{B}}^{2}P_{v_{B}}^{1}\right)}, \end{array}$$
(3)$$\begin{array}{cc} \; & Y_{11}^{X,L}=Y_{11}^{X,f}=\frac{P_{v_{B}}^{1}P_{v_{A}}^{2}Q_{w_{A}w_{B}}+P_{w_{B}}^{1}P_{w_{A}}^{2}P_{v_{A}}^{0}Q_{ow_{B}}+P_{w_{B}}^{1}P_{w_{A}}^{2}P_{v_{B}}^{0}Q_{v_{A}o}}{P_{w_{B}}^{1}P_{v_{B}}^{1}\left(P_{w_{A}}^{1}P_{v_{A}}^{2}-P_{w_{A}}^{2}P_{v_{A}}^{1}\right)}\\ \; & -\frac{P_{w_{B}}^{1}P_{w_{A}}^{2}Q_{v_{A}v_{B}}+P_{w_{B}}^{1}P_{w_{A}}^{2}P_{v_{A}}^{0}P_{v_{B}}^{0}Q_{oo}}{P_{w_{B}}^{1}P_{v_{B}}^{1}\left(P_{w_{A}}^{1}P_{v_{A}}^{2}-P_{w_{A}}^{2}P_{v_{A}}^{1}\right)}-\frac{P_{v_{B}}^{1}P_{v_{A}}^{2}H}{P_{w_{B}}^{1}P_{v_{B}}^{1}\left(P_{w_{A}}^{1}P_{v_{A}}^{2}-P_{w_{A}}^{2}P_{v_{A}}^{1}\right)}, \end{array}$$
(4)$$e_{11}^{X,U}=\frac{T_{w_{A}w_{B}}\left(1+\gamma \sqrt{1/(N_{xw_{A}w_{B}}T_{w_{A}w_{B}})}\right)-H/2}{P_{w_{A}}^{1}P_{w_{B}}^{1}Y_{11}^{X,L}},$$
(5)$$\begin{array}{c} e_{11}^{Z,U}=\frac{T_{s_{A}s_{B}}+P_{s_{A}}^{0}P_{s_{B}}^{0}T_{oo}-\left[P_{s_{A}}^{0}T_{os_{B}}+P_{s_{B}}^{0}T_{s_{A}o}\right]}{P_{s_{A}}^{1}P_{s_{B}}^{1}Y_{11}^{Z,L}}, \end{array}$$
(6)$$H=P_{w_{A}}^{0}Q_{ow_{B}}+P_{w_{B}}^{0}Q_{w_{A}o}-P_{w_{A}}^{0}P_{w_{B}}^{0}Q_{oo},$$
where $P_{l_{A}}^{n}\left(P_{l_{B}}^{m}\right)$ denotes the photon-number distribution of the source at Alice’s (Bob’s) side, $Q_{l_{A}l_{B}}$ and $T_{l_{A}l_{B}}$ are the gain and the total quantum bit errors [25], and H is the combination of the gain of the decoy state and the vacuum state. γ is the standard error, and its value is set to 5.3 here. The expression for $Y_{11}^{X,L}$ is equal to $Y_{11}^{X,e}$ when $P_{v_{A}}^{1}P_{w_{A}}^{2}P_{w_{B}}^{1}P_{v_{B}}^{2}$ < $P_{w_{A}}^{1}P_{v_{A}}^{2}P_{v_{B}}^{1}P_{w_{B}}^{2}$, otherwise the expression equals $Y_{11}^{X,f}$[19,24]. Considering the effect of statistical fluctuations on multiple observations, the method of collective constraints can provide tighter constraint conditions between different sources ($s_{A},w_{A},v_{A},s_{B},w_{B},v_{B},o$) than independent bounds. Thus, these parameters $Y_{11}^{X,L},e_{11}^{Z,U},H$ can be further optimized to achieve a higher key rate by the joint constraints method [8].

By the above formulas, we can calculate the final secret key rate of the asymmetric seven-intensity MDI-QKD protocol.

### 2.2. Asymmetric MDI-QKD with AD

Many previous works have demonstrated that the AD method can further improve the performance of QKD [20,21,22,23]. In this section, we improve the secure key rate and transmission distance of the asymmetric seven-intensity MDI-QKD protocol with the AD method. An additional AD method is performed between parameter estimation and post-processing step, and highly correlated bit pairs are discriminated from weakly correlated information. The security of AD method will be analyzed in an entanglement-based scheme. Alice prepares the quantum state $\frac{1}{\sqrt{2}}\left(|00\rangle +|11\rangle \right)$ and sends the second particle to Bob through the quantum channel. Since Eve controls the quantum channel by certain value $\lambda_{i}$ (i = 0, 1, 2, 3), the quantum state shared between Alice and Bob after transmission can be expressed by the following formula:(7)$$\begin{array}{c} \sigma_{AB}=\lambda_{0}\left|\phi_{0}\rangle \langle \phi_{0}\right|+\lambda_{1}\left|\phi_{1}\rangle \langle \phi_{1}\right|+\lambda_{2}\left|\phi_{2}\rangle \langle \phi_{2}\right|+\lambda_{3}\left|\phi_{3}\rangle \langle \phi_{3}\right|, \end{array}$$
(8)$$\begin{array}{c} |\phi_{0}\rangle =\frac{1}{\sqrt{2}}\left(|00\rangle +|11\rangle \right),\\ |\phi_{1}\rangle =\frac{1}{\sqrt{2}}\left(|00\rangle -|11\rangle \right),\\ |\phi_{2}\rangle =\frac{1}{\sqrt{2}}\left(|01\rangle +|10\rangle \right),\\ |\phi_{3}\rangle =\frac{1}{\sqrt{2}}\left(|01\rangle -|10\rangle \right), \end{array}$$
and $\lambda_{0}+\lambda_{1}+\lambda_{2}+\lambda_{3}=1$. For the quantum state $\sigma_{AB}$, the bit error rate of Alice and Bob’s measurements on different bases can be expressed as $\lambda_{1}+\lambda_{3}=e_{1}^{x}$ (four-state or six-state protocol), $\lambda_{2}+\lambda_{3}=e_{1}^{z}$ (four-state or six-state protocol), and $\lambda_{1}+\lambda_{2}=e_{1}^{y}$ (six-state protocol). Eve can steal information and reduce the key rate by choosing the certain value $\lambda_{i}$ and the secret key rate can be given by [20]:(9)$$\begin{array}{cc} R & \geq min_{\lambda_{0},\lambda_{1},\lambda_{2},\lambda_{3}}\left[H\left(X|E\right)-H\left(X|Y\right)\right]\\ & =min_{\lambda_{0},\lambda_{1},\lambda_{2},\lambda_{3}}\left[1-\left(\lambda_{0}+\lambda_{1}\right)h\left(\frac{\lambda_{0}}{\lambda_{0}+\lambda_{1}}\right)-\left(\lambda_{2}+\lambda_{3}\right)h\left(\frac{\lambda_{2}}{\lambda_{2}+\lambda_{3}}\right)-h\left(\lambda_{0}+\lambda_{1}\right)\right]. \end{array}$$

In the AD method, Alice and Bob divide their own raw bits into blocks $\left(x_{1},\ldots ,x_{b}\right)$ and $\left(y_{1},\ldots ,y_{b}\right)$ of size *b*. Then, choosing a random binary value *c*, Alice sends $\left(x_{1}⊕c,\ldots ,x_{1}⊕c\right)$ to Bob. Bob compares this bitstring with their bitstring $\left(y_{1},\ldots ,y_{b}\right)$ and accepts the security of information only if the results are either all zeros or all ones in the block. In the two cases accepted, Alice (Bob) saves the first bit $x_{1}$$\left(y_{1}\right)$ of the initial string as the raw key. Thus, AD can discern highly correlated bitstring from weakly correlated information as the final raw key. Obviously, the successful probability of the AD method on a certain block of size *b* can be calculated by:(10)$$\begin{array}{c} P_{succ}={\left(\lambda_{0}+\lambda_{1}\right)}^{b}+{\left(\lambda_{2}+\lambda_{3}\right)}^{b}. \end{array}$$

After performing the AD step, the practical QBER value $\lambda_{2}+\lambda_{3}$ in the *Z* basis can be replaced by $\frac{\left(\lambda_{2}+\lambda_{3}\right)}{P_{succ}}$, and the practical QBER in the *X* basis also can be recalculated. The quantum state shared between Alice and Bob can be replaced by:(11)$$\begin{array}{cc} \; & \sigma_{AB}={\bar{\lambda }}_{0}\left|\phi_{0}\rangle \langle \phi_{0}\right|+{\bar{\lambda }}_{1}\left|\phi_{1}\rangle \langle \phi_{1}\right|+{\bar{\lambda }}_{2}\left|\phi_{2}\rangle \langle \phi_{2}\right|+{\bar{\lambda }}_{3}\left|\phi_{3}\rangle \langle \phi_{3}\right|, \end{array}$$
(12)$$\begin{array}{c} {\bar{\lambda }}_{0}=\frac{{\left(\lambda_{0}+\lambda_{1}\right)}^{b}+{\left(\lambda_{0}-\lambda_{1}\right)}^{b}}{2P_{succ}},\\ {\bar{\lambda }}_{1}=\frac{{\left(\lambda_{0}+\lambda_{1}\right)}^{b}-{\left(\lambda_{0}-\lambda_{1}\right)}^{b}}{2P_{succ}},\\ {\bar{\lambda }}_{2}=\frac{{\left(\lambda_{2}+\lambda_{3}\right)}^{b}+{\left(\lambda_{2}-\lambda_{3}\right)}^{b}}{2P_{succ}},\\ {\bar{\lambda }}_{3}=\frac{{\left(\lambda_{2}+\lambda_{3}\right)}^{b}-{\left(\lambda_{2}-\lambda_{3}\right)}^{b}}{2P_{succ}}. \end{array}$$

The QKD protocol enhanced by the AD method can achieve the secret key at rate [20]:(13)$$\begin{array}{c} R\geq max_{b}\;\frac{1}{b}P_{succ}min_{{\bar{\lambda }}_{0},{\bar{\lambda }}_{1},{\bar{\lambda }}_{2},{\bar{\lambda }}_{3}}[1-\left({\bar{\lambda }}_{0}+{\bar{\lambda }}_{1}\right)h\left(\frac{{\bar{\lambda }}_{0}}{{\bar{\lambda }}_{0}+{\bar{\lambda }}_{1}}\right)-\left({\bar{\lambda }}_{2}+{\bar{\lambda }}_{3}\right)h\left(\frac{{\bar{\lambda }}_{2}}{{\bar{\lambda }}_{2}+{\bar{\lambda }}_{3}}\right)\\ -h({\bar{\lambda }}_{0}+{\bar{\lambda }}_{1})]. \end{array}$$

Based on the previous analysis, the AD method can be combined with the QKD protocol. It has been widely used in other protocols in previous works. Similarly, the AD method can be applied to further optimize the properties of quantum channels in the asymmetric MDI-QKD. When the AD method is combined with the asymmetric seven-intensity MDI-QKD protocol, the secret key rate can be estimated by the following formula: (14)$$\begin{array}{c} R\geq P_{s_{A}}P_{s_{B}}\frac{1}{b}q_{succ}Q_{s_{A}s_{B}}^{Z}\{{\left(\frac{P_{11}Y_{11}^{Z,L}}{Q_{s_{A}s_{B}}^{Z}}\right)}^{b}\left[1-\left({\bar{\lambda }}_{0}+{\bar{\lambda }}_{1}\right)h\left(\frac{{\bar{\lambda }}_{0}}{{\bar{\lambda }}_{0}+{\bar{\lambda }}_{1}}\right)-\left({\bar{\lambda }}_{2}+{\bar{\lambda }}_{3}\right)h\left(\frac{{\bar{\lambda }}_{2}}{{\bar{\lambda }}_{2}+{\bar{\lambda }}_{3}}\right)\right]\\ -f_{e}h\left({\bar{E}}_{s_{A}s_{B}}^{Z}\right)\}, \end{array}$$
(15)$$P_{11}=s_{A}e^{-s_{A}}s_{B}e^{-s_{B}},$$
(16)$$\begin{array}{c} q_{succ}={\left(E_{s_{A}s_{B}}^{Z}\right)}^{b}+{\left(1-E_{s_{A}s_{B}}^{Z}\right)}^{b}, \end{array}$$
(17)$$\begin{array}{c} {\bar{E}}_{s_{A}s_{B}}^{Z}=\frac{{\left(E_{s_{A}s_{B}}^{Z}\right)}^{b}}{{\left(E_{s_{A}s_{B}}^{Z}\right)}^{b}+{\left(1-\left(E_{s_{A}s_{B}}^{Z}\right)\right)}^{b}}, \end{array}$$
where $P_{11}$ is the probability of both Alice and Bob’s signal states emitting single-photon events, $q_{succ}$ is the successful probability of the AD method, ${\bar{E}}_{s_{A}s_{B}}^{Z}$ is the error rate value after the AD method, and $e_{11}^{x}$ and $e_{11}^{z}$ are the single-photon error rate in the *X* and *Z* bases, respectively. Note that $Y_{11}^{X,L}$, $e_{11}^{x}$, and $e_{11}^{z}$ can be estimated with the decoy-state method.

## 3. Results

In this work, we explore the combination of a QKD and a post-processing method. We adopt the asymmetric seven-intensity MDI-QKD protocol and the AD method, which can improve the performance of asymmetric MDI-QKD protocol greatly. In this section, numerical simulations of the asymmetric seven-intensity MDI-QKD protocol with AD method are given and the simulation parameters are shown in Table 1. After analyzing the simulation results, we obtained the following significant research results.

We analyze the secret key rate of the asymmetric MDI-QKD protocol with and without the AD method, and the corresponding comparison results are shown in Figure 1 under different conditions $L_{asy}$ = 0 dB, 12 dB, 24 dB. The figure shows that the key rate with and without AD are consistent within a short distance. However, for example, the red line with $L_{asy}$ = 12 dB, the AD method has a clear advantage at a transmission loss of about 33 dB, and a final transmission loss reaching 39 dB as well as the secret key rate showing a clear improvement. For a more obvious exploration of the reason, we present Figure 2 with respect to *b*. We can observe that, in the above example, the value of *b* at about 33 dB has changed from 1 to 2, indicating that the AD method begins to work. With the increase of transmission loss, the AD method requires a larger *b* value to obtain a tight correlation from weak correlation. Furthermore, the results of the above case are similar to the other two cases ($L_{asy}$ = 0 dB, $L_{asy}$ = 24 dB). Therefore, the AD method can improve the key generation rate of asymmetric MDI-QKD over a long distance.

Additionally, we also further investigate the specific effects of the AD method on an asymmetric MDI-QKD under various values $L_{asy}$, and the results are shown in Figure 3. We describe the meaning of Figure 3 and give a detailed definition of the improved percentage. Generally, when the degree of asymmetry is large, the deterioration of the key rate becomes more obvious. However, after the AD method is used, it can be clearly seen in Figure 3 that the improvement effect of the AD method becomes more obvious with the increase of the degree of asymmetry. For example, the improved percentage can reach about 35% when the value $L_{asy}$ = 35 dB, which means that AD method can better solve some transmission performance bottlenecks of the entire network.

By the above analysis, the AD method indeed can increase the propagation distance when the number of pulse pairs $N=10^{11}$. In order to further analyze the finite size effects, we give the simulation results in Figure 4 under different values of *N* when the value $L_{asy}$ = 12 dB. As can be seen from Figure 4, the AD method improves the performance of the asymmetric MDI-QKD protocol under various finite-size effects. Even though there is a large statistical fluctuation when the number of pulse pairs $N=10^{10}$, the AD method can still tolerate transmission losses of more than 5 dB, which means that AD method can also be more adaptable with finite-size cases.

## 4. Conclusions

The AD method, a classical algorithm based on information theory, can be combined with QKD without changing the existing system structure. Specifically, the AD method can be combined with an asymmetric seven-intensity MDI-QKD to improve the robustness effectively, so as to distinguish and extract highly correlated bit pairs from the weakly correlated information as the final secret key. The AD method has a better performance for the asymmetric MDI-QKD protocol. The greater the degree of asymmetry, the better the improvement of the AD method. The AD method can also improve the performance of the asymmetric MDI-QKD protocol under various finite-size effects, and can be more adaptable with finite-size cases. Our work may play a role in measurement-device-independent networks. 

## Figures and Tables

**Figure 1 entropy-25-01174-f001:**
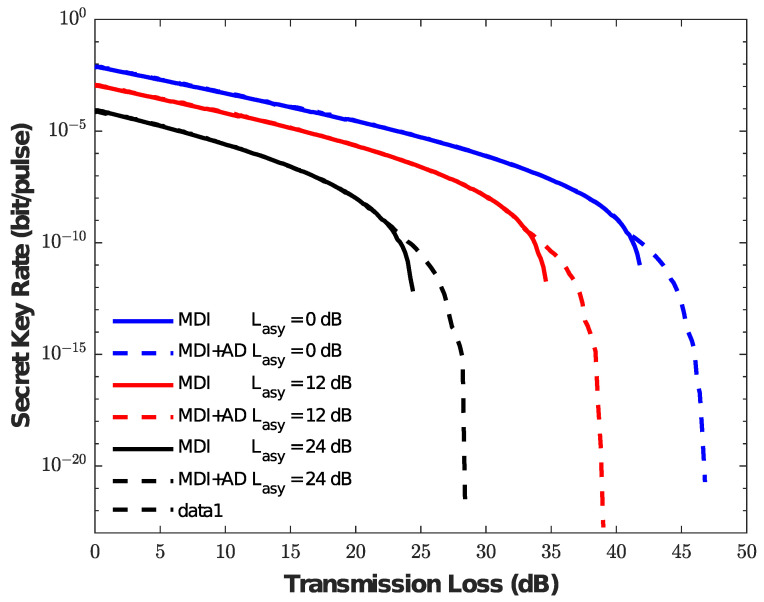
Comparison of the secret key generation rate versus the transmission loss. The value $L_{asy}$ is the loss difference of Alice to Charlie and Bob to Charlie. The different colors represent loss difference, which is $L_{asy}$ = 0 dB, $L_{asy}$ = 12 dB, and $L_{asy}$ = 24 dB, respectively. The solid line represents the secret key without the AD method, and the dotted line represents the secret key with the AD method.

**Figure 2 entropy-25-01174-f002:**
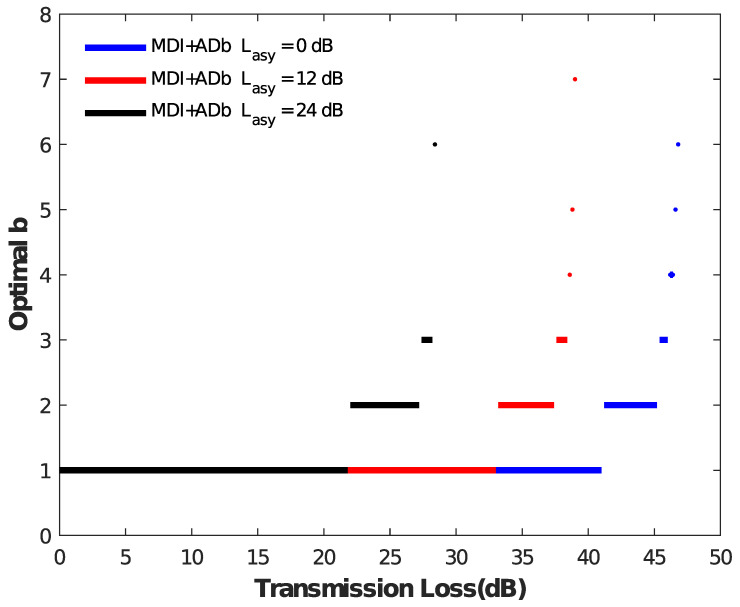
Results of the optimal *b* versus the transmission loss. The black, red, and blue represent the values $L_{asy}$ = 0 dB, $L_{asy}$ = 12 dB, and $L_{asy}$ = 24 dB, respectively. When value *b* is not equal to 1, the AD method can further improve the secret key rate and transmission distance of the asymmetric MDI-QKD protocol.

**Figure 3 entropy-25-01174-f003:**
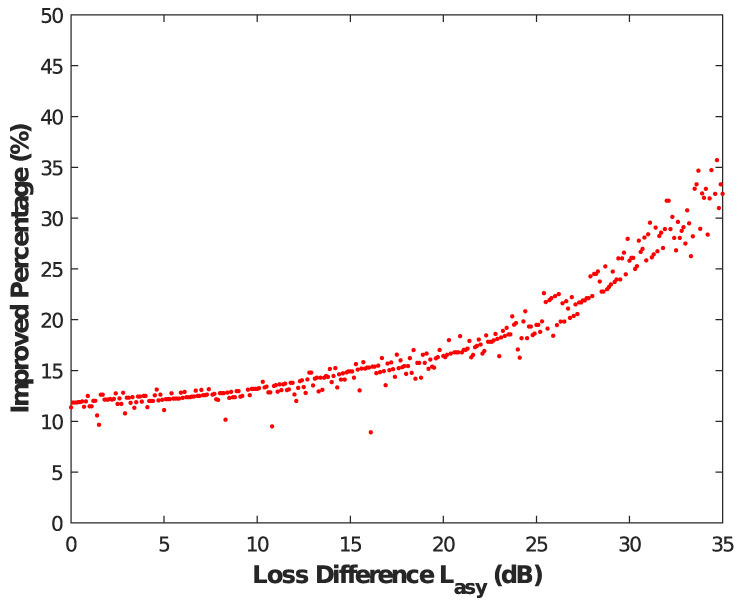
Results of the value $L_{asy}$ versus the improved percentage. The improved transmission loss is the difference of the maximum transmission loss of the asymmetric MDI-QKD with and without the AD method, and we define the improved percentage as the difference divided by the latter. With the increasing degree of asymmetry, the improved percentage also becomes better.

**Figure 4 entropy-25-01174-f004:**
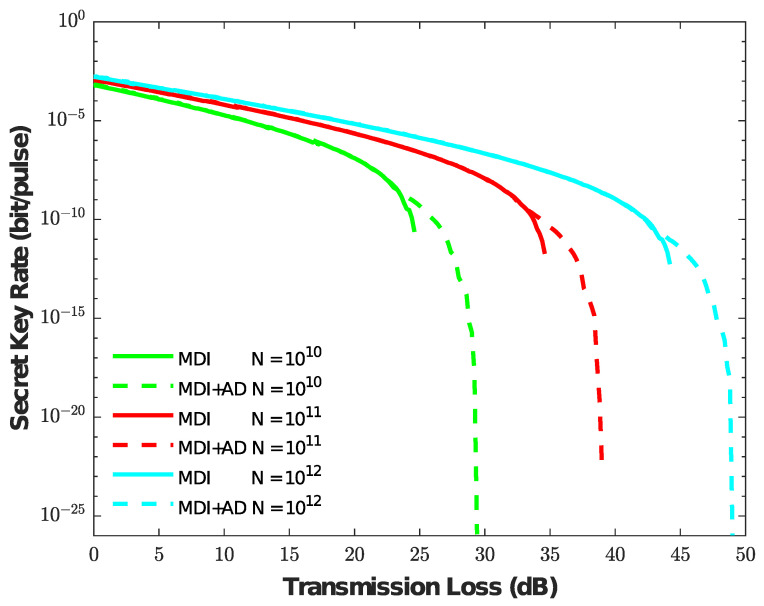
Comparison of the secret key generation rate versus the transmission loss when the value $L_{asy}$ = 12 dB. The different colors represent the values $N=10^{10}$, $N=10^{11}$, and $N=10^{12}$, respectively. The solid line represents the secret key without the AD method, and the dotted line represents the secret key with the AD method.

**Table 1 entropy-25-01174-t001:** The basic system parameters used in our numerical simulations. $\eta_{D}$ and $Y_{0}$ are the efficiency and dark count rate of detectors at Charlie’s side; $e_{d}$: the misalignment error of the QKD system; $f_{e}$: the error correction efficiency; *N*: the number of pulse pairs Alice and Bob send.

$e_{d}$	$\eta_{D}$	$Y_{0}$	$f_{e}$	*N*
0.5%	65%	$8\times 10^{-7}$	1.16	$10^{11}$

## Data Availability

The data that support the findings of this study are available from the corresponding author upon reasonable request.

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
