# Peer review of "Asymmetric Measurement-Device-Independent Quantum Key Distribution through Advantage Distillation"

_entropy, 2023, doi:10.3390/e25081174_

Round 1

Reviewer 1 Report

There are lots of minor issues with the draft, the authors should carefully check the manuscript: 1. Line 96 ‘is’, line 99 ‘control’, line 108 ‘send’, line 110 ‘one’ ‘case’ and so on. 2. Spelling errors, such as: ‘pluse’ 3. The addresses only contain half of the quotes.

Reviewer 2 Report

Authors describe simulated improvement of performance of advantage distillation method used upon the asymmetric MDI-QKD. 

In my opinion asymmetricity is very likely in a realistic scenario: two communicating stations will generally have different quality or even different technology of their qubit detectors, different length and loss/quality of quantum channel. So the research topic of this paper is very relevant.

The paper is well written, concise and scientifically sound. 

The language is problematic to some extent and should be reviewed by a person fluent in English. I will give some examples but there are much more throughout the text.

Line 1. "3 basic theories of quantum mechanics" does not sound right. For example use "3 basic techniques based on quantum mechanics" or similar.

Line 18. You start with enumerating "theories" from "The first". Please keep enumerating "the second" and "the third". For each, start a new paragraph.

Line 39. Please define acronym AD in the text too, not only in the Abstract.

Line 68. Instead "the possibility" use "the probability".

Line 112. Instead of "identify" you may use "discriminate" or "discern".

In the Results section, please instead of kilometers in distance, use loss in dB on the x-axis of all figures and all results. Do nit mention kilometers at all.. This is done in most other works and it is the only proper way, because kilometers mean nothing. In my personal experience loss of 10 yr old fibers can easily reach 0.35 dB/km at 1550 nm and bad splices and interconnects can convert 50 km into barely 5-10 km. Besides, in Fig. 1. L_asy in the figure itself misses the unit, which is probably "km", but as I said, should be dB loss.

Conclusion. AD is not a "classical algorithm based on QKD", it is "a classical algorithm based in information theory" and there is nothing quantum in it, indeed fully classic. You may wish to rectify that.

Should be reviewed by a person fluent in English.
